# Efficacy and mechanism of high-purity HAMCC combined with CGF in promoting the repair of radiation-induced skin and soft tissue damage

Xiaohao Hu[1☉], Anru Liang[2☉], Tongling Zhao[3☉], Yu Ling[1], Yanlin Wei[1], Yuer Zuo[1], Hongmian Li [1,4*]

1 Department of Emergency, The People's Hospital of Guangxi Zhuang Autonomous Region and Research Center of Medical Sciences, Guangxi Academy of Medical Sciences, Nanning, Guangxi, China, 2 Department of Plastic Surgery, The Second Nanning People's Hospital and The third Affiliated Hospital of Guangxi Medical University, Nanning, Guangxi, China, 3 Development Planning and Discipline Construction Centre, Guangxi Medical University, Nanning, Guangxi, China, 4 Department of Plastic and Aesthetic Surgery, The People's Hospital of Guangxi Zhuang Autonomous Region and Research Center of Medical Sciences, Guangxi Academy of Medical Sciences, Nanning, Guangxi, China

☉ These authors contributed equally to this work.
* lihongmian@gxmu.edu.cn

## Abstract

Radiotherapy, a prevalent and effective treatment for various malignancies, often causes collateral damage to normal skin and soft tissues in the irradiated area. To address this, we developed a novel approach combining SVFG-modified adipose-derived high-activity matrix cell clusters (HAMCC) with concentrated growth factors (CGF) to enhance regeneration and repair of radiation-induced skin and soft tissue injuries. Our study included cellular assays, wound healing evaluations, and histological analyses. In the inflammatory wound healing environment, this treatment reduced ROS and 8-oxo-dG expression while increasing four antioxidant-related proteins (Nrf2, HO-1, NQO1, AKR1C1), thereby suppressing oxidative stress and improving wound healing efficiency. Additionally, the HAMCC and CGF combination promoted collagen expression and rearrangement and accelerated angiogenesis. This innovative treatment presents a promising strategy for regenerating and repairing radiation-induced skin and soft tissue damage.

## 1. Introduction

Radiation therapy, a commonly used regimen for malignant tumors, is applied in over 50% of cancer patients and effectively delays tumor progression [1]. However, while targeting tumor cells, it can also harm normal tissues and cells within the radiation field. Despite ongoing advancements in radiation techniques, acute skin reactions occurring within 1–4 weeks remain a significant concern [2].

**Data availability statement:** All relevant data are within the paper and its Supporting Information files.

**Funding:** This study was financially supported by Guangxi Natural Science Foundation (2024GXNSFAA010169, 2023GXNSFDA026035),the National Nature Science Foundation of China(82460449) and the Youth Science Innovation and Entrepreneurship Talent Training Project of Nanning (RC20190206,RC20220108). The above funders participated in the project design and the final review of the manuscript.

**Competing interests:** The authors have declared that no competing interests exist.

Severe cases can result in epidermal detachment, ulceration, and necrosis [3–5]. Currently, 85% to 95% of cancer patients experience varying degrees of skin damage from radiotherapy, significantly affecting quality of life and imposing economic burdens. This may also lead to psychological distress and potential treatment abandonment [6,7]. Thus, timely intervention for radiation-induced skin damage is crucial. Although topical medications and wound dressings are used to enhance skin regeneration, their efficacy remains suboptimal [8,9]. Therefore, there is an urgent need to develop novel treatment strategies for radiation-induced skin damage.

Concentrated platelet products, including platelet-rich plasma (PRP), platelet-rich fibrin (PRF), and concentrated growth factors (CGF) [10,11], are widely used in medicine, particularly in dentistry, plastic surgery, and sports medicine, to promote tissue repair and healing [12–14]. CGF, a third-generation platelet concentrate introduced in 2010, offers advantages over its predecessors. With optimized centrifugal parameters, CGF exhibits a shorter preparation time and a more robust structure compared to PRP. Additionally, CGF shows a higher content of growth factors and a prolonged release time. The fiber matrix, enriched with abundant growth factors, demonstrates enhanced biological effects [15,16].

Adipose-derived stem cells (ADSCs), pluripotent stem cells located in subcutaneous tissue, exhibit a range of beneficial properties. They can differentiate into various cell types, including osteoblasts, adipocytes, and chondrocytes [17–19]. Compared to other types of stem cells, ADSCs can be harvested through minimally invasive liposuction techniques, and adipose tissue is readily available. Furthermore, ADSCs possess immunomodulatory properties, enabling them to mitigate inflammatory responses and facilitate wound repair. In wound healing research, ADSCs demonstrate significant potential by enhancing healing through multiple mechanisms [20,21]. ADSCs secrete various growth factors, including vascular endothelial growth factor (VEGF), Transforming Growth Factor-β (TGF-β), Platelet Derived Growth Factor (PDGF), Epidermal Growth Factor (EGF), Fibroblast Growth Factor (FGF) and multiple anti-inflammatory factors, which promote neovascularization and reduce wound inflammation. Additionally, ADSCs stimulate the proliferation and migration of surrounding cells, including fibroblasts and keratinocytes, further accelerate wound collagen deposition and re-epithelialization. The Stromal Vascular Fraction Gel (SVFG) is a heterogeneous cell population containing numerous ADSCs and various growth factors, exhibiting robust regenerative capabilities [22,23]. The centrifugation parameters for SVFG were optimized, leading to the preparation of a high biological activity HAMCC. Researchers globally have applied CGF or SVFG individually in wound treatment contexts [24,25]. However, studies exploring the synergistic effects of CGF and SVFG on radiation-induced wounds, as well as their mechanisms of action, are notably scarce.

This study aimed to evaluate the therapeutic efficacy of subcutaneously injecting a specific ratio of HAMCC and CGF complex on radiation-induced skin damage and seeks to elucidate the underlying mechanisms, offering new perspectives and methodologies for clinically managing radiation-induced skin injuries.

## 2. Materials and methods

### 2.1 Experimental reagents and instruments

Osteogenic induction solution (Wuxi Puhe Biomedical Technology Co., Ltd.), 4% gelatin, 1% alizarin red (PH4.2), ethanol (Chinese Medicine); Chondrogenic induction solution (Wuxi Puhe Biomedical Technology Co., Ltd.), oil red O staining solution, toluidine blue (Solarbio, pH2.5), CD34 (Invitrogen Catalog # PA5–85917), CD45 (Invitrogen Catalog # 11-0461-82), CD44 (Invitrogen Catalog # MA5–16906), CD105 (Invitrogen Catalog # MA1–19594), CD106 (Abcam Catalog # ab223983), CD73 (Invitrogen Catalog # 11-0739-42), CD90 (Abcam Catalog # ab95812), PDGF-BB ELISA Kit (BioSH 72328), VEGF ELISA Kit (BioSH 72257), FGF-2 Kit (BioSH 72329), IL-8 Kit (BioSH 72066), IL-10 Kit (BioSH 72056), 8-oxo-dG ELISA Kit (Ybio), ROS ELISA Kit (Ybio). Hematoxylin, eosin (both from Shanghai Zhanyun Chemical Co., Ltd.), Masson Staining Kit (Solarbio), neutral gum (Shanghai Yiyang Instrument Co., Ltd., China), 0.1% Triton x-100, 3%$H_2O_2$/PBS, Antigen Repair Solution, DAB Coloring Kit (ZSGB-BIO, ZLI-9018), Total RNA Extraction Kit (Shanghai Yuduo Biotechnology Co., Ltd.), SYBRGreen PCR Kit (Thermo F-415XL), Reverse Transcription Kit (Thermo #K1622) were employed. Biological X-ray irradiation instrument (Panasonic), electrophoresis instrument (BIO-RAD mini protean 3 cell), electroconversion instrument (PS-9, Dalian Jingmai Technology Co., Ltd.), enzyme-linked immunosorbent assay instrument (Thermo MK3), integrated chemiluminescence imaging instrument (ChemiScope 5300 Pro), Real-time detector (ThermoFisher Scientific Co., Ltd., China ABI-7500).

### 2.2 Preparation of HAMCC and identification of adipose-derived stem cells from HAMCC

**2.2.1 Extraction of HAMCC.** SD rats were euthanized using pentobarbital sodium for anesthesia. Approximately 4 mL of subcutaneous adipose tissue was collected from the back, scapula, anterior neck, and groin. The collected fat was placed in an ice-water mixture for 10 minutes. Subsequently, the lower liquid layer was discarded, and the upper fat layer was collected and fragmented into small pieces using scissors. The fat fragments were then placed into two 10 mL syringes connected by a Luhr connector and repeatedly pushed for 1 minute until the fat formed a chylous consistency.

The collected adipose tissue was centrifuged at 1300 g for 3 minutes, discarding the lower liquid portion and retaining the middle chylous adipose tissue. The chylous adipose tissue was then centrifuged at 2000 g for 3 minutes, with the lower liquid portion and the upper yellow transparent oil discarded, and the middle adipose tissue retained to form HAMCC. HAMCC was inoculated into a culture dish, and cell morphology was examined and photographed on days 1, 3, 5, 7, and 9. Osteogenesis, adipogenesis, and chondrogenesis of the cells were induced, and stem cell identification was conducted using flow cytometry.

**2.2.2 Flow cytometry.** Flow cytometry was used to identify the phenotype of cultured ADSCs. Cultured ADSCs were detached from the culture dish and washed twice with PBS. ADSCs (10,000 cells/sample) were then single-stained with purified antibodies in the dark at 4°C: CD34-FITC, CD44-FITC, CD45-FITC, CD49-FITC, CD73-FITC, CD90-FITC, CD105-FITC, and CD106-FITC. After 30 minutes of incubation with the antibodies, the ADSCs were washed twice and detected using a FACSCalibur instrument (BD Biosciences, California, USA). Data were analyzed using Flowjo software (TreeStar, Inc., Ashland, Oregon, USA).

### 2.3 Extraction, preparation, and identification of CGF

The rats were anesthetized with isoflurane gas to induce coma, and a total of 12 mL of cardiac blood was extracted from four rats. Following extraction, 2 mL of peripheral blood was drawn into a sterile anticoagulation tube and immediately centrifuged using a specialized machine (Medifuge CGF MF 200100 Silfradent SRL, Sofia, FC, Italy). The centrifugation procedure included: 30 s acceleration, 2700 rpm for 2 min, 2400 rpm for 4 min, 2700 rpm for 4 min, 3000 rpm for 3 min, and 36 s deceleration to stop [26]. Following centrifugation, the contents separated into three layers: the bottom layer contained red blood cells, the top layer was a transparent supernatant, and the middle layer contained the CGF gel. The primary

activated CGF gel was carefully extracted from the middle layer. Multiple layers of sterile gauze were used to compress the CGF into a stable membrane-like structure. The CGF was then cut into several millimeter fragments in a sterile culture dish and stored at 4°C for future use. The indices of the CGF gel precipitate were assessed on days 1, 3, 5, and 7 according to ELISA kit guidelines.

## 2.4 Establishment of a radiation-induced skin and soft tissue damage model in rats

A total of 48 SD male rats (180g-220g) were randomly sorted into four groups, with 12 rats in each group. To prepare for irradiation, lead shielding was positioned over the animal. The back skin was pulled out through an aperture in the shielding and carefully affixed to the plexiglass platform outside the shielding. The exposed back skin was irradiated 11 cm from the 160 kVp X-ray source for 8 min with 6.3 mA, administering a dose of 40 Gy with a dose rate of 5Gy/min to a 4 cm² of back skin area. Group A received a combined intervention of 0.25ml CGF and 0.25 ml HAMCC. Group B received an intervention of 0.5 ml HAMCC. Group C received an intervention of 0.5 ml CGF. Group D served as the blank control group and received an intervention of 0.5 ml physiological saline. After 24h of irradiation, photos were taken, and the wound surface was observed. According to the assigned groups, subcutaneous injections of the appropriate drugs were administered into the four quadrants of the irradiation site. Subsequently, the changes in the wound surface were monitored and observed for a duration of one to four weeks after the injection. After the 7th, 14th, and 28th day of irradiation, the skin flap at the irradiation center was excised for future experiments, and wound recovery was observed.

In our study, we employed a random number table method to randomly assign animals to different groups. This involved generating a random sequence using a random number table and then allocating the animals to various groups according to the sequence. In each phase of the experiment, we follow the following methods to ensure that we are as randomized as possible:

1. Allocation Phase: The researchers responsible for conducting the experiment and the data analysts were blinded to the group assignment to avoid selection bias. An independent coordinator or a computerized randomization system should have access to the group allocations to ensure the random sequence is generated and applied correctly.

2. Experimental Phase: In a single-blind experimental setup, the caretakers were not aware of the allocation of the groups. The researchers performing the interventions were aware of the group assignments to facilitate appropriate application of treatments or control conditions.

3. Outcome Assessment Phase: Outcome assessors were blinded to the group allocation to prevent observation bias when evaluating the results of the experiment. Only the individuals administering the treatment or intervention might be aware of the group assignments, but they do not participate in outcome assessment.

4. Data Analysis Phase: The statisticians or data analysts continued to be blinded to the group assignment during the initial analysis phase to avoid analysis bias. Upon completion of data collection and primary analysis, the details of group allocation were revealed to the data analysts for a comprehensive assessment of the results.

All animals in each group were intervened at the same time, and after treatment, each rat was housed separately in the same SPF environment in the same cage, thus eliminating the influencing factors of time, environment, and mutual activity among rats.

## 2.5 Ethic statement

Animal experiments were performed under a project license (No. 202111109) granted by the Ethics Committee of the Animal Care & Welfare Committee of Guangxi Medical University, in compliance with the institutional guidelines for the care and use of animals. A protocol was prepared before the study without registration. All animal experiments complied with

the ARRIVE guidelines and were carried out in accordance with the U.K. Animals (Scientific Procedures) Act, 1986 and associated guidelines, EU Directive 2010/63/EU for animal experiments.

SD rats were housed in a specific pathogen-free (SPF) environment with ad libitum access to food and water. To minimize suffering, animals were anesthetized for all procedures via intraperitoneal injection of pentobarbital sodium (50 mg/kg), with anesthetic depth confirmed by pedal withdrawal reflex and maintained with supplementary doses as needed. Efforts to alleviate pain included maintaining body temperature during procedures, close post-operative monitoring for distress, and administering analgesics when indicated. At experimental endpoints, animals were humanely euthanized by an overdose of pentobarbital sodium (150 mg/kg, i.p.), with death confirmed by cessation of vital signs and reflexes.

## 2.6 Sample collection for further testing

The skin tissues were collected on the 7th, 14th, and 28th day after irradiation and intervention, and were stained with H&E to evaluate the changes in each layer of the skin between different groups. The collagen content was observed utilizing Masson staining, and the expression of CD31 in each group was detected by employing immunohistochemistry. Western blot detected the protein content of Nrf2, HO-1, NQO1, and AKR1C1 in each group; RT-PCR was utilized to compare the expression levels of Nrf2, HO-1, NQO1, and AKR1C1 mRNA in each group.

**2.6.1 RT-qPCR.** After extracting total RNA from skin tissues with Trizol, the first-strand cDNA was generated with Advantage® RT-for-PCR kit (Takara, China). Genious 2X SYBR Green Fast qPCR Mix (Abclonal, China) was used for quantitative real-time PCR.

**2.6.2 Western blot.** Skin tissues were lysed using RIPA buffer (Beyotime, China) supplemented with 1/100 PMSF. Protein concentrations were determined with the BCA protein assay kit (BL521A, Biosharp, China). Equal protein aliquots were heated at 100°C for 10 minutes and separated via 10% SDS-PAGE. Proteins were then transferred onto a PVDF membrane (Merck-Millipore) and blocked with 5% BSA. The membrane was incubated overnight with primary antibodies, followed by secondary antibodies for 1 hour. Protein bands were visualized using the ECL substrate kit (WBKLS0100, Merck-Millipore).

**2.6.3 Histological and immunohistochemical staining.** Tissue samples collected on days 7, 14, and 28 post-wounding from the four groups were fixed in 4% paraformaldehyde, embedded in paraffin, sectioned at a thickness of 5 μm, mounted on slides, deparaffinized, rehydrated and stained with H&E as per the instructions. Masson staining was performed using the Trichrome Stain (Masson) Kit (Solarbio, Beijing, China).

In order to explore the regulatory effects of different treatments on wound angiogenesis, immunohistochemical staining was performed using anti-CD31 antibody. After dewaxing and rehydration, antigen retrieval was performed according to the kit manual. Endogenous peroxidase activity was blocked by adding an appropriate blocking agent and incubating at room temperature for 10 minutes. The sections were then rinsed with PBS buffer three times, each for 3 minutes. Based on tissue size, 100 μL or an appropriate volume of primary antibody solution was applied, followed by incubation at 37°C for 60 minutes. The sections were rinsed again with PBS buffer three times, each for 3 minutes. Subsequently, 100 μL or an appropriate amount of reaction enhancement solution was added and incubated at 37°C for 20 minutes, followed by three PBS rinses. An enhanced enzyme-labeled goat anti-mouse IgG polymer (100 μL or appropriate volume) was then applied and incubated at 37°C for 20 minutes, followed by three PBS rinses. DAB color development was performed by adding an appropriate amount of freshly prepared DAB solution, incubating at room temperature for 5–8 minutes. The sections were rinsed with tap water and stained with hematoxylin for 20 seconds, followed by differentiation, washing, and bluing. Finally, the sections were dehydrated, cleared, and mounted. Post-staining, the tissue sections were examined under a microscope to interpret the results.

**2.6.4 Tunel assay.** Cell apoptosis in skin wound tissue subjected to different interventions was assessed using the terminal deoxynucleotidyl transferase-mediated dUTP nick end-labeling (TUNEL) assay. Briefly, sections were processed with the TUNEL Detection Kit (KGA7033, KeyGEN BioTECH, China) according to the manufacturer's instructions. The

sections were counterstained with hematoxylin for histological orientation and observed using the inverted fluorescence microscope (OLYMPUS, JAPAN).

## 2.7 Statistical analysis

All statistical analyses were performed using IBM SPSS Statistics v22.0. For the one-way ANOVA, the data were first assessed for normality using the Shapiro-Wilk test. If the data were normally distributed, a one-way ANOVA was conducted to compare the means across different groups. If the data were not normally distributed, the Kruskal-Wallis test was used as a non-parametric alternative. Post-hoc comparisons were made using Tukey's HSD test for normally distributed data or the Mann-Whitney U test for non-normally distributed data to determine which groups were significantly different from each other. For the two-way ANOVA, the data were first checked for normality and homogeneity of variances using the Shapiro-Wilk test and Levene's test, respectively. If the assumptions were met, a two-way ANOVA was performed to assess the main effects of each factor and their interaction. Post-hoc comparisons were conducted using Tukey's HSD test for normally distributed data or the Mann-Whitney U test for non-normally distributed data.Effect sizes were calculated using Cohen's d for continuous variables to provide an indication of the magnitude of the differences between groups. The threshold for small, medium, and large effects was set at 0.2, 0.5, and 0.8, respectively. All statistical tests were set at a significance level of $p < 0.05$, and post-hoc adjustments were made using Bonferroni correction to control for multiple comparisons.

## 3. Results

### 3.1 Identification of CGF and differentiation ability of ADSCs from HAMCC

After 24–48 hours of primary cell culture, the cells adhered to the bottom of the culture dish, predominantly exhibiting a wide and flattened morphology, with several triangular or polygonal cells. As the culture period extended, parietal cells formed colonies of different sizes, and a characteristic spindle cell morphology became evident (Fig 1A).

The outcomes of flow cytometry indicated that the positive expression rates of CD73, CD44, CD90, CD105, and CD106 exceeded 90%, while the positive expression rates of CD49 and CD45 were below 10% (Fig 1B).

Following a two-week induction of HAMCC osteogenesis, alizarin red staining revealed the presence of numerous mineral nodules. Similarly, after a two-week induction, oil red O staining demonstrated the accumulation of abundant lipid droplets. After induction for two weeks, toluidine blue staining demonstrated a significant number of positive expressions (Fig 1C–1E).

The CGF gel precipitates at different time points were collected for ELISA detection. The results showed that IL-8 continued to decrease, and IL-10 concentration continued to increase from the 1st to the 7th day. Moreover, the concentrations of FGF-2 and VEGF continued to increase slowly, and the concentration of PDGF-BB gradually decreased before the 5th day and significantly increased after the 7th day (Fig 2).

### 3.2 CGF+HAMCC accelerated wound healing after irradiation

40Gy was selected for subsequent formal experiments. The stages of radiation-induced wounds were evaluated using a skin damage scoring system, based on the acute radiation murine flank reaction criteria established by Korpela et al [27]. The wounds reached maximal median damage scores between days 7–14, then the scores dropped by day 21, and the wound continues to heal until time of sacrifice on day 28. The CGF+HAMCC group exhibited the minimal wound damage compared to the other groups, with the most favorable recovery observed by 28th day. (Fig 3A, 3B).

### 3.3 Improvement of epidermis regeneration capacity and collagen deposition in the wound after CGF+HAMCC intervention

The HE staining results showed that on day 7, the control group exhibited severe epidermal damage, with minimal inflammatory cell infiltration. In contrast, the CGF group, HAMCC group, and CGF+HAMCC group showed no significant infiltration of inflammatory cells (Fig 4A, 4B).

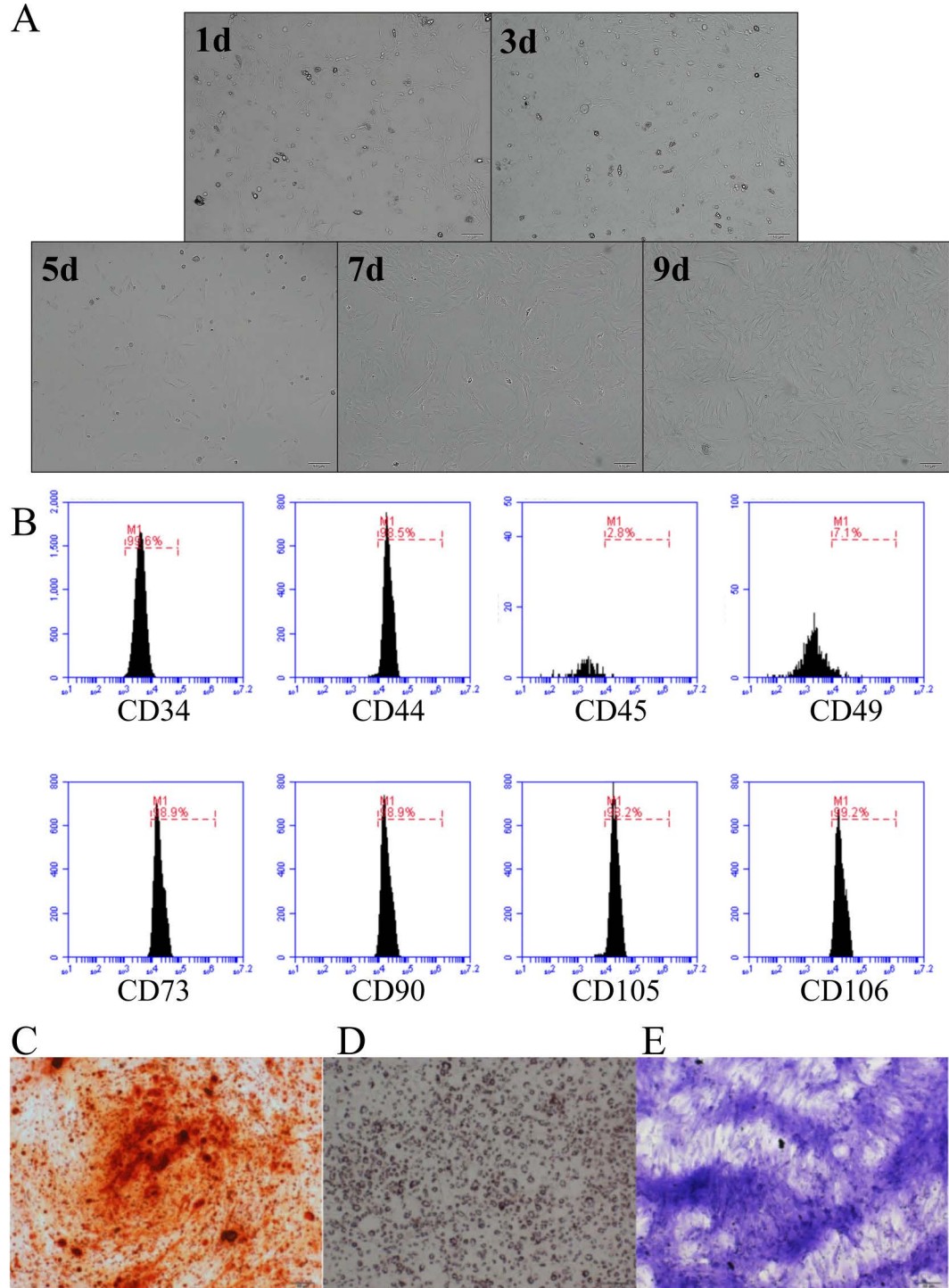

**Fig 1. Identification of stem cell components in HAMCC. (A)** Cell morphology of HAMCC in a culture dish. After two weeks of HAMCC culture, osteogenic induction. **(B)** Detection of stem cell components in HAMCC by flow cytometry. **(C)** Adipogenic induction, **(D)** Osteogenic induction and (E) chondrogenic induction were performed to detect stem cell differentiation capacity.

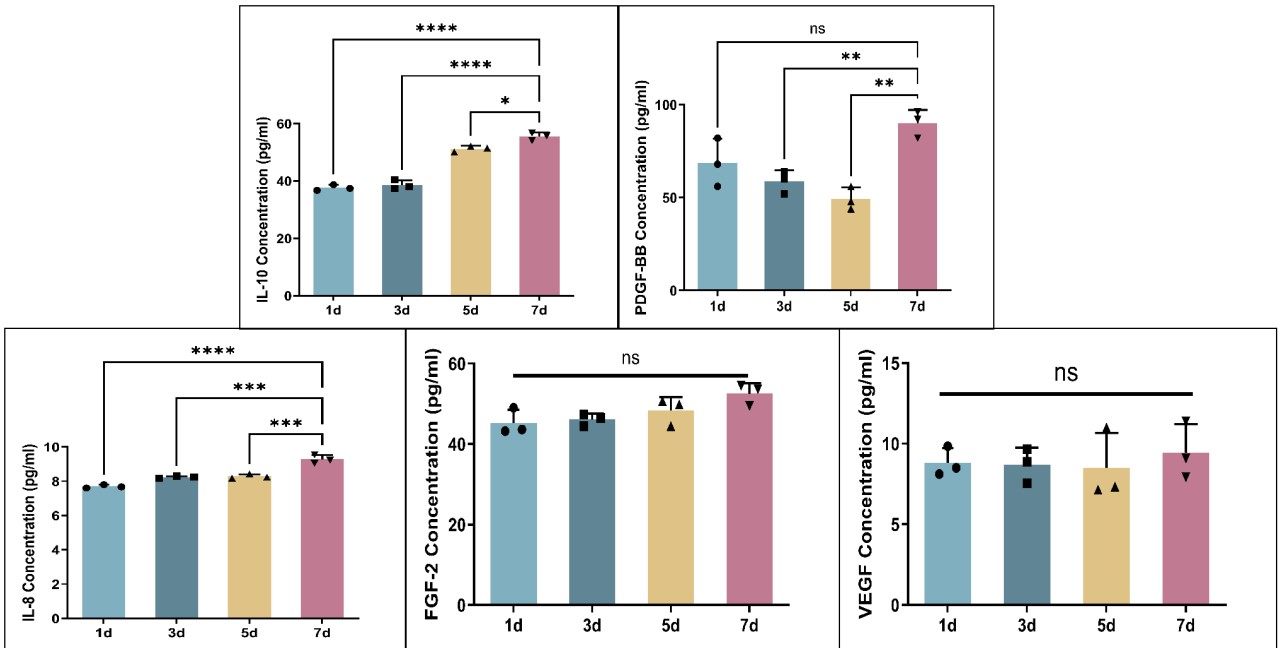

**Fig 2. Identification of effective components of CGF.** Detection of cytokines in CGF by ELISA has shown that CGF can continuously secrete many kinds of cytokines such as for a period of time. *P<0.05, **P<0.01, ***P<0.001, ****P<0.0001.

On day 14, the control group indicated remarkable epidermal thickening and necrosis, disrupted dermal architecture, absence of prominent red blood cell exudation and substantial inflammatory cell infiltration. The CGF and HAMCC groups showed partial epidermal recovery and some dermal structural disarray. In the CGF+HAMCC group, the epidermis largely recovered, with only minor dermal structural irregularities. On day 28, the CGF+HAMCC group presented partial epidermal thickening, a nearly normal dermal structure, no significant red blood cell infiltration, and limited inflammatory cell presence compared to other groups.

Masson staining results revealed a significant increase in collagen content in the CGF+HAMCC group compared to the control group. On day 14, the collagen content in CGF+HAMCC group was the highest, while there was no significant difference in collagen content in the HAMCC group and CGF groups relative to that of the control group. By day 28, the CGF+HAMCC group showed a remarkable increase in collagen content compared to the other groups, indicating that CGF+HAMCC intervention significantly accelerated deposition. (Fig 4C, 4D). (*P<0.05, **P<0.01, in contrast to the blank control group; #P<0.05, ##P<0.01, in contrast to the CGF intervention group; &P<0.05, &&P<0.01, compared to the HAMCC intervention group)

### 3.4 CGF+HAMCC reduced cell apoptosis and promoted angiogenesis in vivo

The result of the Tunel test showed that on the 14th day after irradiation, the proportion of apoptotic cells in the CGF+HAMCC group, HAMCC group and CGF group was significantly lower compared to the control group, in which the CGF+HAMCC group showed the lowest ratio of apoptotic cells. On the 14th day, the proportion of apoptotic cells in all groups decreased significantly, indicating that the skin tissues were significantly healed compared to day 14. The HAMCC group still maintained a low proportion of apoptosis, but it had no statistically difference from other groups. (Fig 5A, 5B). (*P<0.05,**P<0.01, compared to the blank control group; #P<0.05, ##P<0.01, in contrast to the CGF intervention group; &P<0.05, &&P<0.01, in contrast to the HAMCC intervention group)

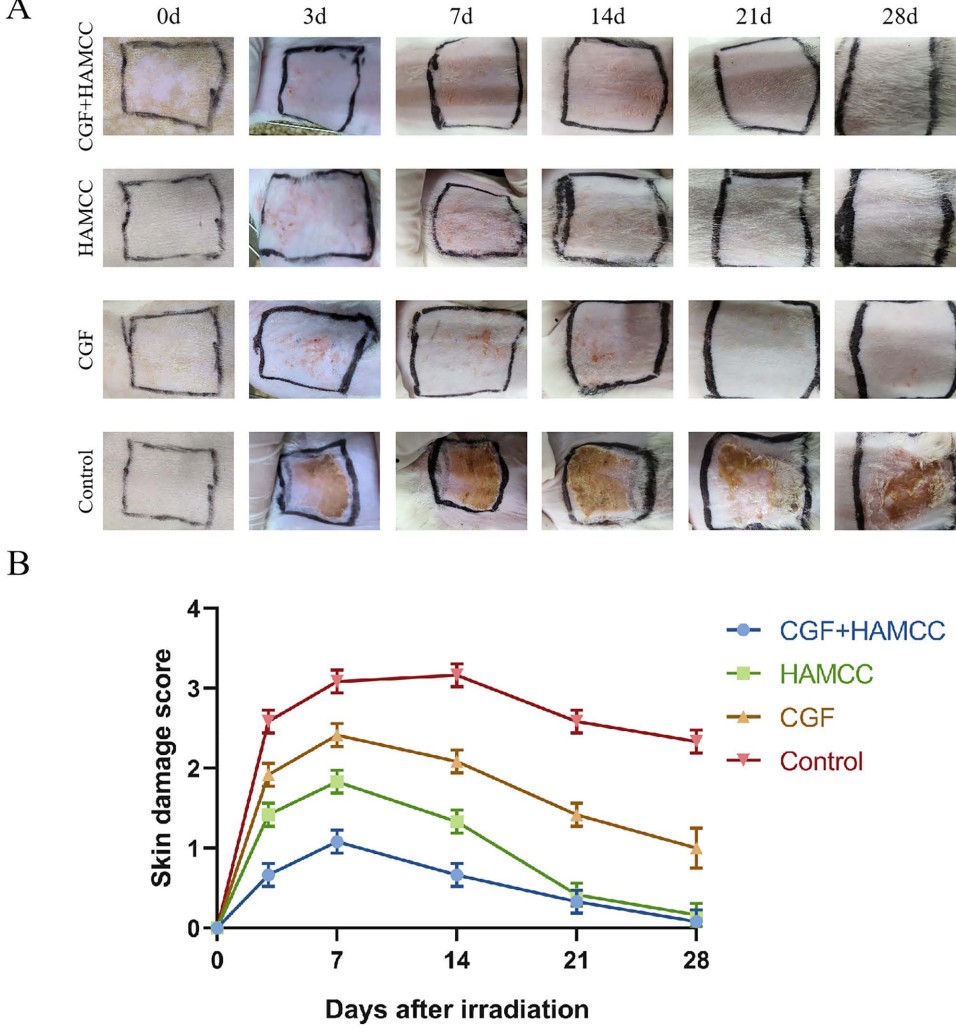

**Fig 3. HAMCC promotes healing of irradiated wounds.** CGF combined with HAMCC significantly promotes wound healing. **(A)** Images of wounds after CGF+HAMCC, HAMCC, CGF, and Normal saline (NS) treatments. **(B)** Radiation skin damage scores of rats exposed to radiation over time. Healing rate (%) compared with the wound area on day 0.

The immunohistochemistry staining showed that the CD31 expression levels of the CGF+HAMCC group and HAMCC group were obviously increased compared to the control group. On the 7th and 14th day after irradiation, the CD31 expression in the CGF+HAMCC group, HAMCC group and CGF group was increased. However, the difference was not significant compared to the blank control group. On the 28th day, the CD31 expression in the CGF+HAMCC group and HAMCC group were significantly higher than the control group, and CGF+HAMCC group showed the highest level. (Fig 5C, 5D). (*$P < 0.05$, **$P < 0.01$, compared to the blank control group, #$P < 0.05$, ##$P < 0.01$, in contrast to the CGF intervention group, &$P < 0.05$, &&$P < 0.01$, in contrast to the HAMCC intervention group)

### 3.5 CGF+HAMCC reduced oxidative stress

The ELISA outcomes indicated that on the 7th day post-treatment, there were no significant differences in indicators among the groups. By the 14th day, levels of ROS and 8-oxo-dG in the radiation combined intervention group and the

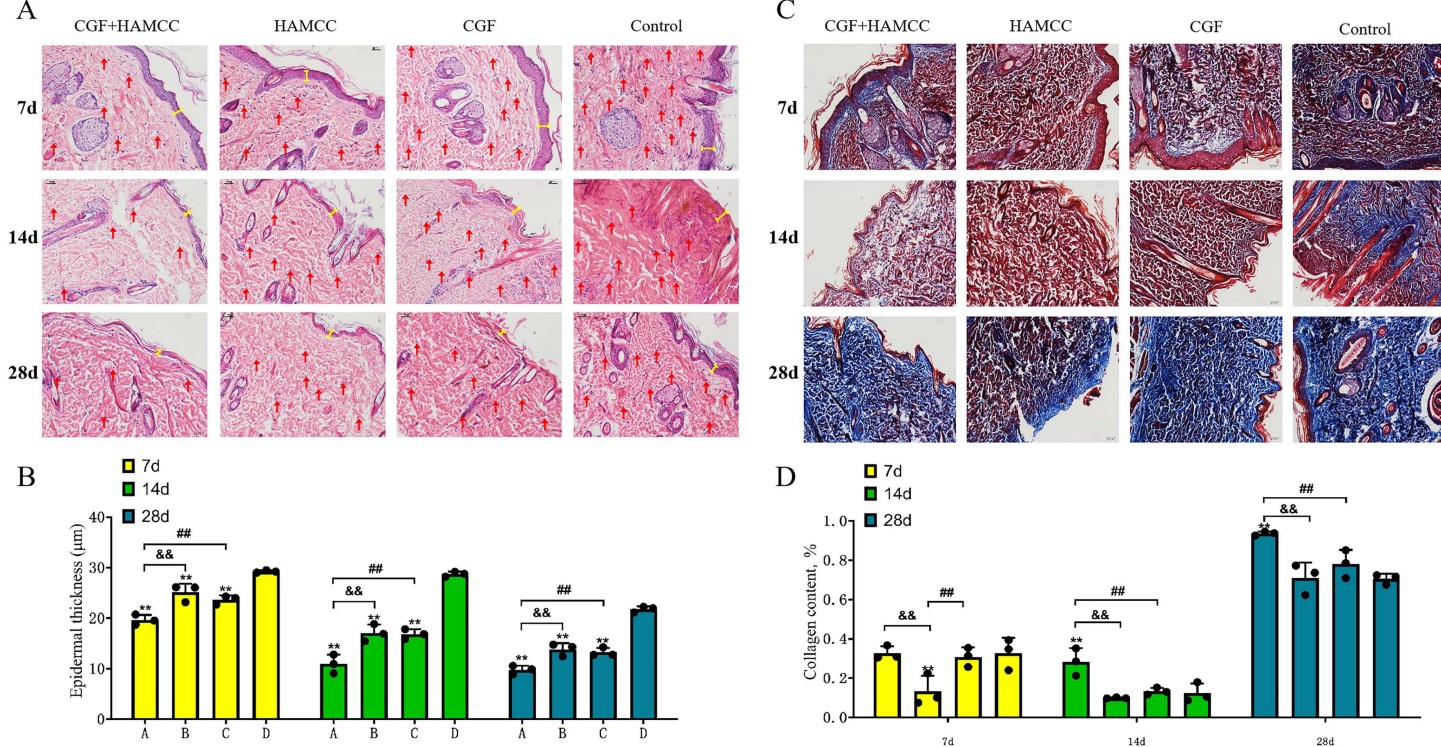

**Fig 4. HAMCC optimizes wound recovery and accelerates collagen regeneration. (A, B)** Representative images of hematoxylin and eosin (H&E) staining of wound tissue. The red arrows represent proinflammatory cells. The epidermal thickness was indicated by a yellow line and quantified. **(C, D)** Masson staining was used to detect the promoting effect of CGF and HAMCC on collagen expression in the wound, and the results were quantified. Group A: CGF+HAMCC; Group B: HAMCC; Group C: CGF; Group D: NS. *$P < 0.05$, **$P < 0.01$,vs. control group; #$P < 0.05$, ##$P < 0.01$, vs. CGF group; &$P < 0.05$, &&$P < 0.01$, vs. HAMCC group.

HAMCC intervention group were significantly reduced compared to the blank control group. The 8-oxo-dG levels in the CGF intervention group were notably decreased relative to the blank control group, while ROS levels showed no significant difference, although they were lower than those in the blank control group. The ROS and 8-oxo-dG levels in the blank control group peaked on the 28th day (Fig 6).

Western Blot and RT-PCR results demonstrated that the expression levels of Nrf2, HO-1, NQO1, and AKR1C1 in the combined intervention group were significantly elevated compared to the blank control group and the CGF intervention group at various time points post-irradiation. On the 14th and 28th days, levels of Nrf2, HO-1, NQO1, and AKR1C1 in the HAMCC intervention group were considerably higher than those in the blank control group; there was no significant variation in mRNA and protein expression between the CGF intervention group and the blank control group (Fig 7). (*$P < 0.05$, **$P < 0.01$, in contrast to the blank control group; #$P < 0.05$, ##$P < 0.01$, in contrast to the CGF intervention group; &$P < 0.05$, &&$P < 0.01$, in contrast to the HAMCC intervention group)

## 4. Discussion

Over half of cancer patients will undergo radiation therapy, with wound healing in irradiated skin presenting a significant clinical challenge. Many patients experience radiation toxicity due to the absorption of radiation by tissues surrounding the tumor target area [28]. Currently, no clinically approved drugs or treatments are available that are both well-tolerated and

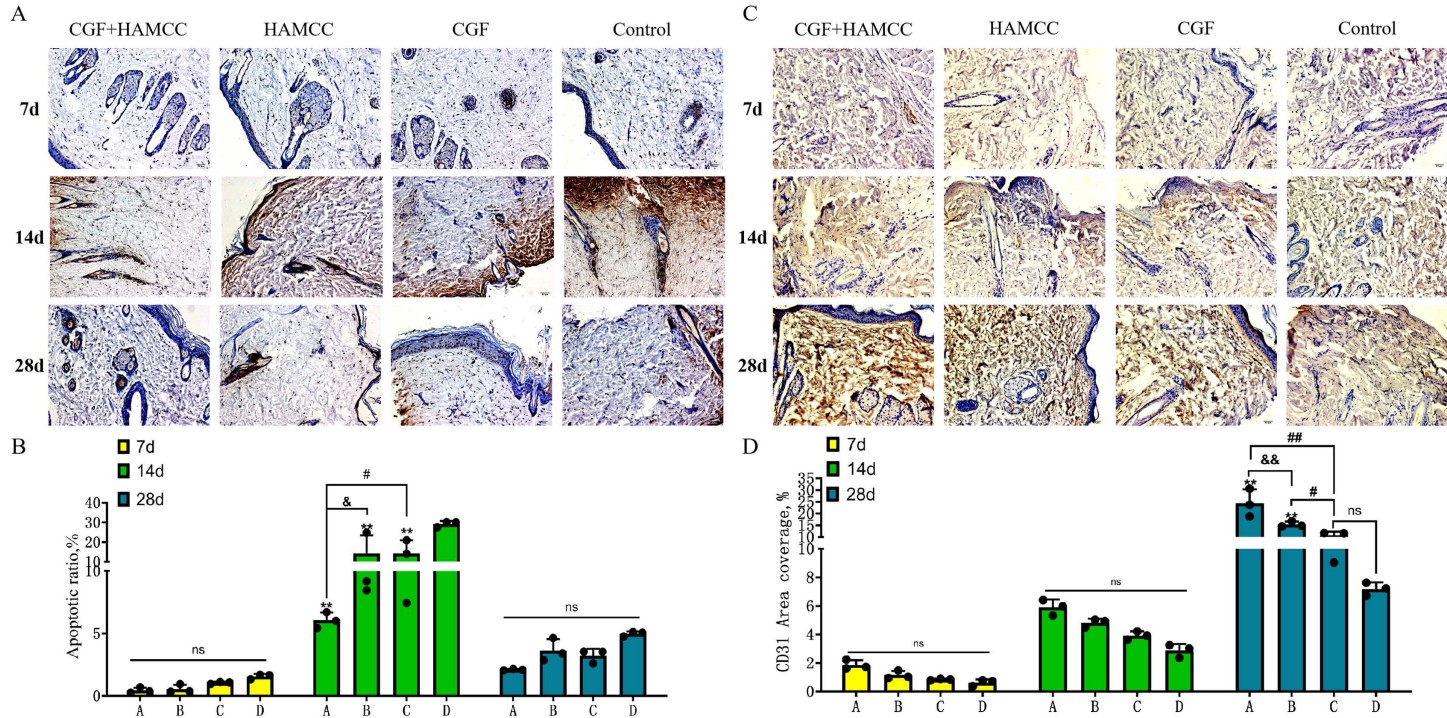

**Fig 5. HAMCC reduces apoptosis and promotes angiogenesis in wound sites. (A, B)** Inhibition of apoptosis by HAMCC detected by Tunel and quantified. **(C, D)** Immunohistochemical staining of CD31 expression in tissues. Group A: CGF+HAMCC; Group B: HAMCC; Group C: CGF; Group D: NS. *$P<0.05$, **$P<0.01$, vs. control group; #$P<0.05$, ##$P<0.01$, vs. CGF group; &$P<0.05$, &&$P<0.01$, vs. HAMCC group.

effective in preventing or mitigating radiation toxicity in normal tissues. Previous studies have demonstrated that platelet derivatives and adipose-derived stromal components can significantly enhance the healing of various skin wounds [24,29].

CGFs are a new generation of platelet derivatives, abundant in fibrin and growth factors, which promote cell proliferation and differentiation [30]. Recently, the application of CGF in wound repair has garnered significant attention. Research has demonstrated that CGF can enhance wound healing, reduce scar formation, and improve the quality of wound repair [31]. Additionally, CGF exhibits various biological functions, including antibacterial, anti-inflammatory, and antioxidant properties, effectively promoting diverse physiological processes during wound healing [32,33].

Adipose-derived stem cells (ADSCs) have attracted significant research interest due to their numerous advantages since their discovery. Compared to traditional bone marrow mesenchymal stem cells, ADSCs are easier to extract and can be obtained in larger quantities with minimal bodily harm [34]. Additionally, ADSCs exhibit low immunogenicity, lack ethical concerns, possess potential for multidirectional differentiation, and secrete various cytokines. These attributes have facilitated their widespread application in tissue repair therapy and regenerative medicine, offering promising prospects [35–37].

SVFG is a heterogeneous cell population extracted from adipose tissue, primarily including endothelial cells, interstitial cells and pericytes, immune cells [38]. SVFG is closely related to ADSCs, with ADSCs being a crucial component. Compared to using ADSCs alone, SVFG offers a simpler preparation process, typically isolated from adipose tissue through mechanical centrifugation or enzyme digestion, eliminating the need for complex cell culture and amplification. Furthermore, SVFG encompasses a richer variety of cell types, including ADSCs and various cells beneficial for tissue repair and regeneration, which can synergistically promote biological activities such as tissue repair, anti-inflammatory responses,

A

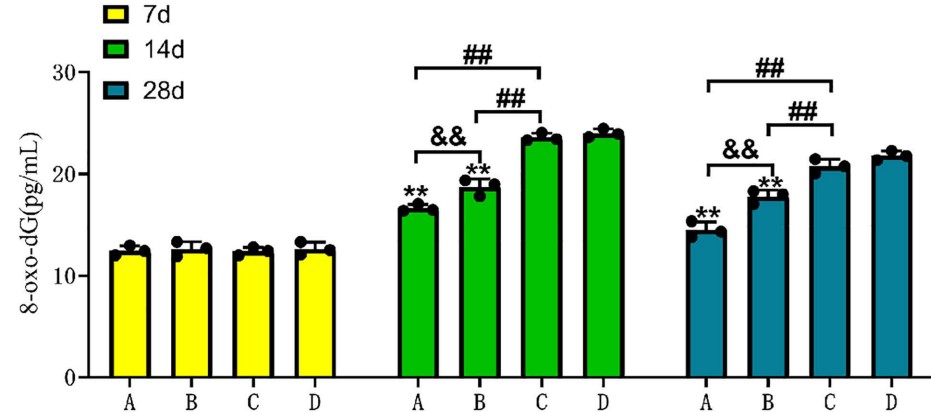

B

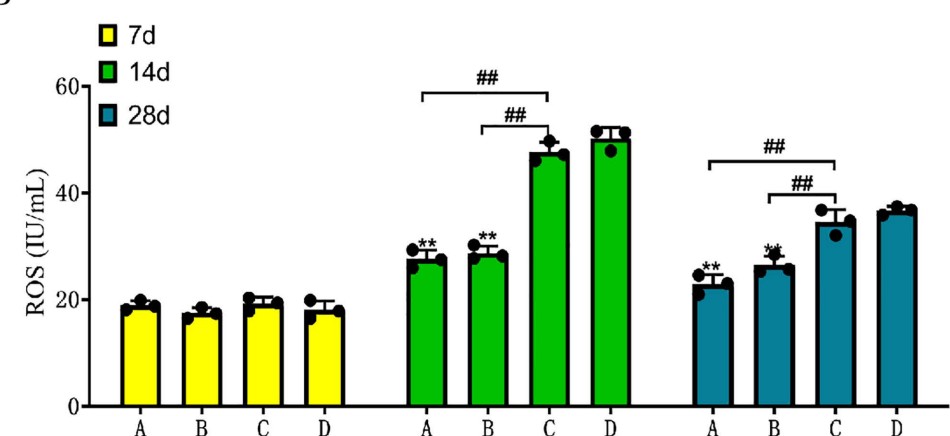

**Fig 6. HAMCC significantly reduces oxidative stress in wounds.** ELISA was used to detect and quantify 8-oxo-dG (A) and ROS (B) levels of each group on the 7th,14th and 28th day. Group A: CGF+HAMCC; Group B: HAMCC; Group C: CGF; Group D: NS. *$P < 0.05$, **$P < 0.01$, vs. control group; #$P < 0.05$, ##$P < 0.01$, vs. CGF group; &$P < 0.05$, &&$P < 0.01$, vs. HAMCC group.

and angiogenesis [39,40]. Based on SVFG, centrifugation parameters were further optimized to obtain a cell cluster with enhanced biological activity—High Activity Matrix Cell Cluster (HAMCC).

In this experiment, HAMCC and CGF were isolated and prepared from rats. These components were then combined to treat radiation-induced skin and soft tissue damage. The findings revealed a significantly higher wound healing rate in the group receiving the combined HAMCC and CGF treatment compared to other treatment groups and the control group. HE staining results further indicated that the combined intervention group exhibited excellent epidermal recovery and a more orderly dermal structure, suggesting that HAMCC combined with CGF significantly enhances skin healing and reconstruction following radiation damage. Collagen, a major component of the dermis, plays a critical role in skin wound healing. The content and arrangement of collagen significantly affect the quality and speed of healing. Research indicates that increased collagen content correlates with faster healing [41]. Masson staining outcomes demonstrated that collagen content in wounds treated with HAMCC and CGF was significantly higher than in other groups. This finding, along with HE staining results, suggests that HAMCC combined with CGF effectively promotes collagen synthesis and rearrangement.

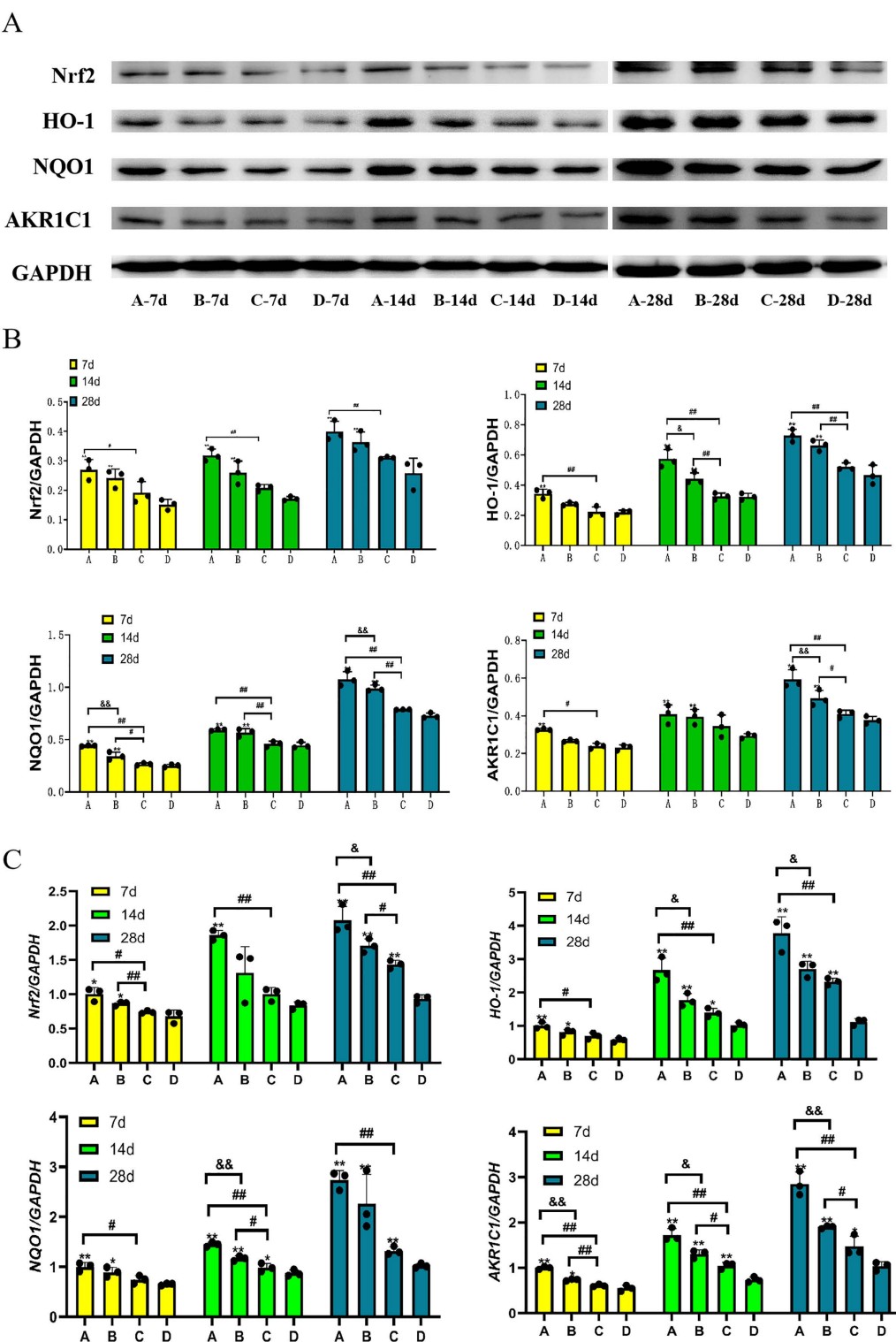

**Fig 7. HAMCC significantly upregulates the expression of antioxidant proteins. (A, B)** Western blot was conducted for the respective wound samples to detect the expression level of each kind of protein. **(C)** The mRNA expression levels of target genes in the wound following treatment. Group A: CGF+HAMCC; Group B: HAMCC; Group C: CGF; Group D: NS. *$P<0.05$, **$P<0.01$, vs. control group; #$P<0.05$, ##$P<0.01$, vs. CGF group; &$P<0.05$, &&$P<0.01$, vs. HAMCC group.

Reactive oxygen species (ROS) are a class of highly active oxidizing substances, including superoxide anions ($O_2^-$), hydroxyl radicals (-OH), hydrogen peroxide ($H_2O_2$), etc. Excessive ROS could cause oxidative stress and damage cell membranes, DNA, proteins, and other biological macromolecules, leading to various diseases. 8-oxo-dG was an oxidative damage product in DNA and was one of the markers of DNA oxidative damage caused by ROS [42].

The NRF2/antioxidant response element (ARE) signaling pathway is a central mechanism by which cells respond to oxidative stress [43]. NRF2, upon activation, translocates to the nucleus and binds to AREs in the promoters of antioxidant genes, leading to their transcription and subsequent expression. In the context of radiation-induced skin and soft tissue damage, the activation of the NRF2 signaling pathway by CGF and HAMCC could be particularly beneficial. Radiation therapy, while effective against cancer cells, often causes collateral damage to normal tissues, leading to oxidative stress and subsequent tissue injury [44]. By upregulating NRF2 and its downstream enzymes, CGF and HAMCC may help protect normal tissues from radiation-induced oxidative damage, thereby promoting healing and reducing the severity of radiation dermatitis.

The ELISA results from our study provide compelling evidence that CGF and HAMCC effectively reduce oxidative stress by upregulating the NRF-2 pathway. The significant decrease in ROS and 8-oxo-dG levels observed in the treated groups suggests that these treatments enhance the cellular antioxidant response. By activating NRF-2, CGF and HAMCC stimulate the expression of key antioxidant enzymes, such as HO-1 and NQO1, which are pivotal in scavenging reactive oxygen species and repairing oxidative damage. This upregulation of NRF-2 and its downstream targets not only mitigates oxidative stress but also contributes to the overall healing process by creating a more favorable cellular environment for tissue repair.

NRF-2 is known to modulate macrophage polarization, with its activation often leading to a shift towards the M2 phenotype, which is characterized by anti-inflammatory and tissue repair activities [45]. Given that oxidative stress can promote the M1 phenotype, which is pro-inflammatory, the decrease in oxidative stress markers in our study could imply a reduction in M1 macrophage activity and an increase in M2 macrophage polarization.

It is speculative to suggest that CGF and HAMCC might be influencing macrophage polarization through the NRF-2 pathway. The reduction in ROS and 8-oxo-dG levels could be indicative of a more favorable microenvironment that promotes M2 polarization, thereby facilitating tissue repair. However, without direct evidence of macrophage polarization, this remains a hypothesis that requires further investigation.

CD31 is a marker of neovascularization [46]. Immunohistochemical studies revealed that CD31 expression in the skin was significantly elevated following HAMCC combined with CGF treatment, suggesting that this combined approach promotes neovascularization. In summary, the current findings indicate that the efficacy of HAMCC combined with CGF spans multiple stages of radiation-induced skin wounds—such as inflammation, proliferation, and remodeling—thereby enhancing the overall repair of radiation-induced skin soft tissue damage.

However, this study has certain limitations that warrant consideration. While our findings demonstrate the therapeutic efficacy of HAMCC combined with CGF, a key limitation is the lack of in vivo cell tracking. We did not assess the long-term survival, migration, or specific localization of the transplanted HAMCC within the irradiated skin and soft tissue. This omission prevents a more precise understanding of the cellular fate and the direct contribution of HAMCC to the observed regenerative processes, such as their potential to directly differentiate into tissue-specific cells or their sustained paracrine effects at the wound site. Future studies will aim to address this by employing advanced cell labeling and imaging techniques to track the transplanted cells in vivo, thereby providing a more comprehensive elucidation of their mechanistic roles. Additionally, as noted, a more detailed investigation into macrophage polarization, which is closely associated with the inflammatory phase, was not conducted, and will be a focus of future research.

## 5. Conclusion

In summary, the study results demonstrate that HAMCC combined with CGF significantly enhances wound healing. In the inflammatory environment of wound healing, this combination reduces ROS and 8-oxo-dG expression levels

while promoting the levels of four antioxidant-related proteins—Nrf2, HO-1, NQO1, and AKR1C1—thereby suppressing oxidative stress and improving healing efficiency. Additionally, HAMCC combined with CGF promotes collagen expression and rearrangement and accelerates angiogenesis, facilitating granulation tissue formation during the proliferation phase and collagen deposition during the remodeling phase. Thus, HAMCC and CGF together offer a novel treatment approach, providing new insights for the regeneration and repair of radiation-induced skin and soft tissue damage.

## Supporting information

**S1 File. Identificatioin of HAMCC.**
(ZIP)

**S2 File. Identification of the efficient component in CGF.**
(ZIP)

**S3 File. Animal experiment.**
(ZIP)

**S4 File. HE staining.**
(ZIP)

**S5 File. Masson staning.**
(ZIP)

**S6 File. Tunel.**
(ZIP)

**S7 File. Immunohistochemistry.**
(ZIP)

**S8 File. ELISA.**
(ZIP)

**S9 File. Western Blot.**
(ZIP)

**S10 File. QPCR.**
(ZIP)

## Author contributions

**Conceptualization:** Hongmian Li.

**Data curation:** Xiaohao Hu, Anru Liang.

**Formal analysis:** Xiaohao Hu, Yu Ling.

**Investigation:** Yuer Zuo.

**Methodology:** Xiaohao Hu.

**Resources:** Tongling Zhao.

**Software:** Xiaohao Hu.

**Supervision:** Hongmian Li.

**Validation:** Xiaohao Hu, Hongmian Li.

**Visualization:** Xiaohao Hu.

**Writing – original draft:** Xiaohao Hu, Yanlin Wei, Hongmian Li.

**Writing – review & editing:** Hongmian Li.

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
