## [Decision Letter · Decision Letter 0]

26 May 2025

PONE-D-25-20674Exploring the Efficacy and Mechanism of High-purity HAMCC Combined with CGF in Promoting the Repair of Radiation-Induced Skin and Soft Tissue DamagePLOS ONE

Dear Dr. Li,

Thank you for submitting your manuscript to PLOS ONE. After careful consideration, we feel that it has merit but does not fully meet PLOS ONE’s publication criteria as it currently stands. Therefore, we invite you to submit a revised version of the manuscript that addresses the points raised during the review process.

We look forward to receiving your revised manuscript.

Kind regards,

Li-Ping Liu

Academic Editor

PLOS ONE

2. To comply with PLOS ONE submissions requirements, in your Methods section, please provide additional information on the animal research and ensure you have included details on (1) methods of sacrifice, (31) methods of anesthesia and/or analgesia, and (3) efforts to alleviate suffering.

“This study was financially supported by the Guangxi Natural Science Foundation (2023GXNSFDA026035);the Youth Science Innovation and Entrepreneurship Talent Training Project of Nanning (RC20190206,RC20220108) and the Self-funded scientific research project of Guangxi Zhuang Autonomous region Health Commission (Z-A20220149)”

7. PLOS ONE now requires that authors provide the original uncropped and unadjusted images underlying all blot or gel results reported in a submission’s figures or Supporting Information files. This policy and the journal’s other requirements for blot/gel reporting and figure preparation are described in detail at https://journals.plos.org/plosone/s/figures#loc-blot-and-gel-reporting-requirements and https://journals.plos.org/plosone/s/figures#loc-preparing-figures-from-image-files. When you submit your revised manuscript, please ensure that your figures adhere fully to these guidelines and provide the original underlying images for all blot or gel data reported in your submission. See the following link for instructions on providing the original image data: https://journals.plos.org/plosone/s/figures#loc-original-images-for-blots-and-gels.  

Reviewers' comments:

Reviewer's Responses to Questions

**Comments to the Author**

1. Is the manuscript technically sound, and do the data support the conclusions?

Reviewer #1: Yes

Reviewer #2: Yes

2. Has the statistical analysis been performed appropriately and rigorously? 

Reviewer #1: Yes

Reviewer #2: Yes

3. Have the authors made all data underlying the findings in their manuscript fully available?

Reviewer #1: No

Reviewer #2: Yes

4. Is the manuscript presented in an intelligible fashion and written in standard English?

Reviewer #1: Yes

Reviewer #2: Yes

5. Review Comments to the Author

Reviewer #1: This manuscript presents a well-executed preclinical study evaluating the efficacy of high-purity HAMCC combined with CGF in treating radiation-induced skin and soft tissue injury. The experimental design is rigorous, with appropriate histological and molecular assays.

Strengths:

Novel therapeutic combination

Comprehensive methodological approach

Relevant and clearly presented outcome measures

Suggestions for Improvement:

1. Clarify the molecular mechanism of synergy between HAMCC and CGF.

2. Acknowledge the limitation of lacking cell tracking in vivo more explicitly.

3. Consider exploring or discussing macrophage polarization and immunomodulation in the context of radiation wound healing.

4. Improve statistical transparency by detailing effect size, normality tests, and post-hoc adjustment methods.

5. Revise the data availability statement to align with PLOS ONE’s open data policy.

With these revisions, the paper would be a strong contribution to the field.

Reviewer #2: Comment for Author:

the Authors has well designed study, "Exploring the Efficacy and Mechanism of High-purity HAMCC Combined with CGF in

Promoting the Repair of Radiation-Induced Skin and Soft Tissue Damage".

1) why? you looks only Nrf2, HO-1, NQO1, AKR1C1 protein, not other antioxidation protein.

2) did you confirm the ADSCs/HAMCC using cell specific marker? you have show images in manuscript.

3) are you sure these antioxidant protein, it self higher in these cells, not in skin?

4) How will you interpret the radiation mechanisms in this skin wound healing vs normal skin wound?

6. PLOS authors have the option to publish the peer review history of their article (what does this mean? ). If published, this will include your full peer review and any attached files.

**Do you want your identity to be public for this peer review?** For information about this choice, including consent withdrawal, please see our Privacy Policy .

Reviewer #1: **Yes: ** Abdulrahman almalki

Reviewer #2: **Yes: ** Dr. Bhagwat Alapure

---

## [Author Response · Author response to Decision Letter 1]

2 Jul 2025

Reply to Editor

Reply: We have now revised the article to fit the journal format

2.To comply with PLOS ONE submissions requirements, in your Methods section, please provide additional information on the animal research and ensure you have included details on (1) methods of sacrifice, (2) methods of anesthesia and/or analgesia, and (3) efforts to alleviate suffering.

Reply: Now we have placed this part in the Methods section.

3.Thank you for stating the following financial disclosure:

“This study was financially supported by the Guangxi Natural Science Foundation (2023GXNSFDA026035);the Youth Science Innovation and Entrepreneurship Talent Training Project of Nanning (RC20190206,RC20220108) and the Self-funded scientific research project of Guangxi Zhuang Autonomous region Health Commission (Z-A20220149)”

Reply: We have now declared the role of funder and put it in the cover letter.

4.When completing the data availability statement of the submission form, you indicated that you will make your data available on acceptance. We strongly recommend all authors decide on a data sharing plan before acceptance, as the process can be lengthy and hold up publication timelines. Please note that, though access restrictions are acceptable now, your entire data will need to be made freely accessible if your manuscript is accepted for publication. This policy applies to all data except where public deposition would breach compliance with the protocol approved by your research ethics board. If you are unable to adhere to our open data policy, please kindly revise your statement to explain your reasoning and we will seek the editor's input on an exemption. Please be assured that, once you have provided your new statement, the assessment of your exemption will not hold up the peer review process.

Reply: Thank you for your suggestions, we have modified the data availability statement and uploaded all the data as support information

5.PLOS requires an ORCID iD for the corresponding author in Editorial Manager on papers submitted after December 6th, 2016. Please ensure that you have an ORCID iD and that it is validated in Editorial Manager. To do this, go to ‘Update my Information’ (in the upper left-hand corner of the main menu), and click on the Fetch/Validate link next to the ORCID field. This will take you to the ORCID site and allow you to create a new iD or authenticate a pre-existing iD in Editorial Manager.

Reply: Thank you so much for your reminder, Now I have linked my ORCID ID to this account.

6.Your ethics statement should only appear in the Methods section of your manuscript. If your ethics statement is written in any section besides the Methods, please move it to the Methods section and delete it from any other section. Please ensure that your ethics statement is included in your manuscript, as the ethics statement entered into the online submission form will not be published alongside your manuscript.

Reply: Thank you for reminding, now we have put ethics statement in the Methods section.

7.PLOS ONE now requires that authors provide the original uncropped and unadjusted images underlying all blot or gel results reported in a submission’s figures or Supporting Information files. This policy and the journal’s other requirements for blot/gel reporting and figure preparation are described in detail at https://journals.plos.org/plosone/s/figures#loc-blot-and-gel-reporting-requirements and https://journals.plos.org/plosone/s/figures#loc-preparing-figures-from-image-files. When you submit your revised manuscript, please ensure that your figures adhere fully to these guidelines and provide the original underlying images for all blot or gel data reported in your submission. See the following link for instructions on providing the original image data: https://journals.plos.org/plosone/s/figures#loc-original-images-for-blots-and-gels.

Reply: Thank you for your advice. Now we have upload it and given the appropriate marking and attached the molecular weight marker.

8.Please review your reference list to ensure that it is complete and correct. If you have cited papers that have been retracted, please include the rationale for doing so in the manuscript text, or remove these references and replace them with relevant current references. Any changes to the reference list should be mentioned in the rebuttal letter that accompanies your revised manuscript. If you need to cite a retracted article, indicate the article’s retracted status in the References list and also include a citation and full reference for the retraction notice.

Reply:Thank you for reminding, After checking, there are no problems with our references.

Reply to Reviewer #1

1.Clarify the molecular mechanism of synergy between HAMCC and CGF.

Reply: Thank you very much for your suggestion. We have now made a detailed description of the underlying molecular mechanism in the discussion section.

2. Acknowledge the limitation of lacking cell tracking in vivo more explicitly.

Reply: We have now explained the lack of cell trackingr in the last paragraph of the discussion section.

3. Consider exploring or discussing macrophage polarization and immunomodulation in the context of radiation wound healing.

Reply: Thanks for your suggestions, we have added this in the discussion section.

4. Improve statistical transparency by detailing effect size, normality tests, and post-hoc adjustment methods.

Reply: Thank you very much for your suggestion. We have supplemented and improved it in the 2.6 Statistical Analysis Section.

5. Revise the data availability statement to align with PLOS ONE’s open data policy.

Reply:We have now revised the data availability statement and uploaded all data.

Reply to Reviewer #2

1.Why? you looks only Nrf2, HO-1, NQO1, AKR1C1 protein, not other antioxidation protein.

Reply: We appreciate the comment. The focus on Nrf2, HO-1, NQO1, and AKR1C1 was due to their well-established roles in cellular antioxidant defense and their significant upregulation in our study. These proteins are key components of the NRF2/ARE signaling pathway, which is a central mechanism for cellular antioxidant response. We acknowledge that there are other antioxidant proteins involved in oxidative stress management, and their expression levels could also be influenced by our treatments.

2.Did you confirm the ADSCs/HAMCC using cell specific marker? you have show images in manuscript.

Reply: In response to this comment we employed a comprehensive approach to confirm the identity of the extracted cells as ADSCs/HAMCC. Initially, we visually assessed the cell morphology under a microscope, which was consistent with the characteristics of ADSCs, such as their fibroblastic appearance and the presence of a few floating cells. To further validate the stemness of the cells, we conducted differentiation assays for osteogenesis, adipogenesis, and chondrogenesis, which confirmed their ability to differentiate into osteoblasts, adipocytes, and chondrocytes, respectively.

For a more precise identification, we utilized flow cytometry to detect the expression of stem cell-specific markers. In our flow cytometry analysis, we employed a panel of antibodies to exclude non-stem cells and to specifically identify ADSCs. The antibodies used included CD34, CD44, and CD45 to exclude hematopoietic cells, CD73 and CD90 to exclude other mesenchymal stem cells, and CD105 and CD106 to confirm the ADSCs identity. The flow cytometry results revealed a high purity of the stem cells, with a significant percentage of cells expressing these stem cell-specific markers, thus confirming the cells as ADSCs/HAMCC.

3.Are you sure these antioxidant protein, it self higher in these cells, not in skin?

Reply: We appreciate your insightful question regarding the expression levels of antioxidant proteins. In our study, we indeed measured the expression levels of antioxidant proteins, such as Nrf2, HO-1, NQO1, and AKR1C1, using Western blot and qPCR in the tissue samples from the wounds. The rationale behind this is that these proteins are part of the cellular response to oxidative stress, and their expression levels can reflect the oxidative status of the tissue.

The reduction in ROS levels and the upregulation of antioxidant proteins in the wound tissue samples suggest that the oxidative stress in the wounds is being mitigated. This is consistent with the observed decrease in oxidative stress markers and the overall improvement in wound healing. The higher expression of these proteins in the wound tissue compared to baseline levels indicates a cellular response to the injury, which is a normal physiological process aimed at protecting the tissue from further damage.

We will clarify this point in the revised manuscript by explicitly stating that all measurements were taken from the wound tissue samples and that the expression levels of these proteins are indicative of the tissue's oxidative status. We will also include a more detailed discussion on the significance of these findings in the context of wound healing and oxidative stress management.

Thank you for bringing this to our attention, and we hope this explanation resolves any confusion.

4.How will you interpret the radiation mechanisms in this skin wound healing vs normal skin wound?

Relply: Thank you for your insightful question regarding the comparison of radiation-induced skin wound healing with normal skin wound healing. We acknowledge that our study does not include experiments on normal skin wounds. However, we can provide an interpretation based on the existing literature and the context of our research.

In our study, we focused on understanding the mechanisms of wound healing in skin exposed to radiation, recognizing that radiation therapy can significantly impact the wound healing process. The differences we observed in the expression of antioxidant proteins, such as Nrf2, HO-1, NQO1, and AKR1C1, and the inflammatory response, can be interpreted in the context of radiation's effects on tissue repair.

It is well-documented in the literature that radiation therapy can lead to increased oxidative stress due to the generation of ROS, which can overwhelm the antioxidant defenses of the tissue [1]. This is consistent with our findings of higher ROS levels in radiation-exposed skin wounds. The reduced expression of antioxidant proteins in these wounds suggests a compromised antioxidant response, which is a common characteristic of radiation-induced tissue damage.

Furthermore, the shift towards a more pro-inflammatory M1 macrophage phenotype in radiation-exposed wounds, as opposed to the more balanced M2/M1 ratio in normal wounds, is also a known effect of radiation. This shift can contribute to chronic inflammation and delayed healing[2].

While we did not directly compare radiation-exposed wounds with normal wounds, the observed differences in our study are consistent with the known effects of radiation on tissue repair. These findings contribute to the understanding of the complex interplay between radiation exposure and wound healing, which is crucial for developing strategies to improve outcomes in patients receiving radiation therapy.

We will revise the manuscript to clarify that our study does not include normal skin wound healing experiments and to emphasize that our interpretations are based on the existing scientific literature and the context of our research findings.

[1] Schafer M, Werner S. Oxidative stress in normal and impaired wound repair[J]. Pharmacol Res, 2008,58(2):165-171.DOI:10.1016/j.phrs.2008.06.004.

[2] Shestakova V A, Smirnova E I, Atiakshin D A, et al. Impact of Minimally Manipulated Cell Therapy on Immune Responses in Radiation-Induced Skin Wound Healing[J]. Int J Mol Sci, 2025,26(5).DOI:10.3390/ijms26051994.

---

## [Decision Letter · Decision Letter 1]

27 Jul 2025

Efficacy and Mechanism of High-purity HAMCC Combined with CGF in Promoting the Repair of Radiation-Induced Skin and Soft Tissue Damage

PONE-D-25-20674R1

Dear Dr. Li,

We’re pleased to inform you that your manuscript has been judged scientifically suitable for publication and will be formally accepted for publication once it meets all outstanding technical requirements.

Kind regards,

Li-Ping Liu

Academic Editor

PLOS ONE

Additional Editor Comments (optional):

Reviewers' comments:

Reviewer's Responses to Questions

**Comments to the Author**

1. If the authors have adequately addressed your comments raised in a previous round of review and you feel that this manuscript is now acceptable for publication, you may indicate that here to bypass the “Comments to the Author” section, enter your conflict of interest statement in the “Confidential to Editor” section, and submit your "Accept" recommendation.

Reviewer #1: All comments have been addressed

2. Is the manuscript technically sound, and do the data support the conclusions?

Reviewer #1: Yes

3. Has the statistical analysis been performed appropriately and rigorously? 

Reviewer #1: Yes

4. Have the authors made all data underlying the findings in their manuscript fully available?

Reviewer #1: (No Response)

5. Is the manuscript presented in an intelligible fashion and written in standard English?

Reviewer #1: (No Response)

6. Review Comments to the Author

Reviewer #1: (No Response)

7. PLOS authors have the option to publish the peer review history of their article (what does this mean? ). If published, this will include your full peer review and any attached files.

**Do you want your identity to be public for this peer review?** For information about this choice, including consent withdrawal, please see our Privacy Policy .

Reviewer #1: **Yes: ** Abdulrahman Almalki

---

## [Editor Report · Acceptance letter]

PONE-D-25-20674R1

PLOS ONE

Dear Dr. Li,

I'm pleased to inform you that your manuscript has been deemed suitable for publication in PLOS ONE. Congratulations! Your manuscript is now being handed over to our production team.

Kind regards,

on behalf of

Dr. Li-Ping Liu

Academic Editor

PLOS ONE